

# Viral recombination blurs taxonomic lines: examination of single-stranded DNA viruses in a wastewater treatment plant

Victoria M. Pearson[1],  S. Brian Caudle[2] and  Darin R. Rokyta[1]

[1] Department of Biological Science, Florida State University, Tallahassee, FL, USA
[2] Division of Food Safety, Florida Department of Agriculture and Consumer Services, Tallahassee, FL, USA

## ABSTRACT

Understanding the structure and dynamics of microbial communities, especially those of economic concern, is of paramount importance to maintaining healthy and efficient microbial communities at agricultural sites and large industrial cultures, including bioprocessors. Wastewater treatment plants are large bioprocessors which receive water from multiple sources, becoming reservoirs for the collection of many viral families that infect a broad range of hosts. To examine this complex collection of viruses, full-length genomes of circular ssDNA viruses were isolated from a wastewater treatment facility using a combination of sucrose-gradient size selection and rolling-circle amplification and sequenced on an Illumina MiSeq. Single-stranded DNA viruses are among the least understood groups of microbial pathogens due to genomic biases and culturing difficulties, particularly compared to the larger, more often studied dsDNA viruses. However, the group contains several notable well-studied examples, including agricultural pathogens which infect both livestock and crops (*Circoviridae* and *Geminiviridae*), and model organisms for genetics and evolution studies (*Microviridae*). Examination of the collected viral DNA provided evidence for 83 unique genotypic groupings, which were genetically dissimilar to known viral types and exhibited broad diversity within the community. Furthermore, although these genomes express similarities to known viral families, such as *Circoviridae*, *Geminiviridae*, and *Microviridae*, many are so divergent that they may represent new taxonomic groups. This study demonstrated the efficacy of the protocol for separating bacteria and large viruses from the sought after ssDNA viruses and the ability to use this protocol to obtain an in-depth analysis of the diversity within this group.

**Submitted** 10 June 2016
**Accepted** 19 September 2016
**Published** 18 October 2016

Corresponding author
Darin R. Rokyta,
drokyta@bio.fsu.edu,
drokyta@gmail.com

## INTRODUCTION

The majority of genomic diversity is contained in viral genomes, yet only a fraction of this diversity has been described (*Hatfull, 2008*; *Thurber, 2009*; *Clokie et al., 2011*). By filling in gaps on these abundant pathogens, we will be better equipped to handle

future outbreaks. Advanced knowledge of the natural mutational landscape, along with other viral genes present in the environment for potential recombination, could provide background information should related viral genotypes cause outbreaks (*Hambly & Suttle, 2005*). Viral genomes have already provided us with many genes that we can use for our benefit, including capsid genes for drug delivery systems and bacteriophage lysis genes as antibiotics (*Kovacs et al., 2007*; *Schmitz, Schuch & Fischetti, 2010*). Increasing our knowledge of their diversity is necessary to discover new ways in which we can use their genes and gene products (*Hatfull, 2008*; *Godzik, 2011*).

Culture-based studies are important to fully understand viruses that can be grown *in vitro*. Over 100 years of development has decreased the cost of conducting culture-based studies while vastly improving the culturing methods available (*Leland & Ginocchio, 2007*). Additionally, approval of these techniques by multiple regulatory boards has resulted in broad implementation. However, we can only culture an extremely limited range of viruses and must complement these approaches with the development and implementation of culture-independent methods (*Rosario & Breitbart, 2011*). Additionally, standard Sanger-sequencing for monitoring new viral pathogens is limited as new viral variants cannot be amplified via PCR until appropriate amplification primers have been created. Therefore, it is necessary to explore viral diversity using new sequencing technologies that do not require the ability to culture the viruses or design specific primers (*Thurber, 2009*). Culture-independent sequencing methods can improve the monitoring of pathogens in public-health facilities by allowing the sequencing of viral variants that cannot be cultured or amplified via PCR until appropriate primers are available and by decreasing the wait time (*Svraka et al., 2010*; *Barzon et al., 2011*; *Schmieder & Edwards, 2012*).

The ssDNA group of viruses is comprised of seven described viral families: *Anelloviridae, Circoviridae, Geminiviridae, Inoviridae, Microviridae, Nanoviridae,* and *Parvoviridae* (*King, Adams & Lefkowitz, 2012*). The ssDNA viruses have mutation rates of $10^{-4} - 10^{-7}$/nucleotide site/generation, similar to RNA viruses, and likewise are small, with genomes ranging from roughly 2–6 kb (*Holmes, 2010*; *Benson et al., 2005*). Members of this group include common agricultural pests, such as porcine circoviruses 1 and 2, maize streak virus, and beet curly top virus (*Fauquet, 2006*). Historically, those involved with agriculture were most interested in increasing our knowledge of this group of viruses, however it is now evident that these viruses are pervasive in many environments and therefore deserve more attention (*Rosario, Duffy & Breitbart, 2012*).

We sought to uncover members of the three known ssDNA families with circular genomes, *Circoviridae, Geminiviridae*, and *Microviridae*, using a protocol biased towards such genomes. The *Circoviridae* family have genomes ranging from 1.7–4 kb (*Benson et al., 2005*; *King, Adams & Lefkowitz, 2012*). All known members of this family are animal viruses, and the majority infect birds. Their genome encodes capsid and replication genes and occasionally 1–2 accessory proteins (*Cheung, 2012*). These genes may be arranged ambisense, with the genes radiating out from the origin in either direction, or unidirectionally (*Benson et al., 2005*). *Geminiviridae* have genomes sizes that range from 2.5–3.1 kb, which is within the range of *Circoviridae* genomes. All known members are

plant viruses with bi-directional genomes and 4–6 genes; some members are monopartite and contain four genes and some are bipartite and have six genes split between two capsids (*Bradeen, Timmermans & Messing, 1997*). Recombination has previously been observed between *Circoviridae* and *Geminiviridae* (*Roux et al., 2013*). Both the *Circoviridae* and *Geminiviridae* families have capsid and replication initiation genes that are conserved within the viral families and a variable number of accessory genes, which are not conserved. The *Microviridae* family is comprised of four subfamilies: the Gokushovirinae, the Bullavirinae (formally Microvirinae) (*Śliwa-Dominiak et al., 2013*; *Krupovic et al., 2016*), the Pichovirinae (*Roux et al., 2012*), and the Alpavirinae (*Krupovic & Forterre, 2011*). Bullavirinae have been thoroughly described and are a model system for genetics and evolution studies (*Crill, Wichman & Bull, 1999*; *Wichman et al., 1999*; *Pearson, Miller & Rokyta, 2012*; *Caudle, Miller & Rokyta, 2014*). Members of this family are bacteriophage with genomes sizes from 4.4–6 kb. Gokushovirinae have genome sizes from 4.4–4.9 kb and are comprised of nine genes, whereas Bullavirinae have 11 genes and genomes that range from 5.4–6 kb (*Fane et al., 2006*). Gokushovirinae produce a major capsid protein, which is similar to the F capsid protein produced by the Bullavirinae. Likewise, the two subfamilies use similar replication initiation proteins, gene A in Bullavirinae. The similarities between these genes among these two subfamilies, along with the conserved nature of these genes in the *Circoviridae* and *Geminiviridae*, provide support for using these two genes as the major source of comparison for this study.

Wastewater treatment plants (WWTP) are large bioprocessors that receive influent from households, businesses, and occasionally city street drains, making them reservoirs for plant, animal, and microbial viruses from local (environmental run-off) and distant (plant viruses passed through humans) sources (*Cantalupo et al., 2011*; *Kim et al., 2011*; *Tamaki et al., 2012*). This feature, collecting viruses from multiple sources in the environment, makes WWTPs excellent sites for the study of viral diversity, allowing the investigation of many different viromes with a single collection. However, WWTPs do not just collect viruses, they have the potential to release new viruses into the surrounding area as the effluent from WWTPs have been found to contain large numbers of viruses (*Rosario et al., 2009*). Additionally, the sludge produced during the treatment of wastewater is often used to fertilize nearby fields or sent to landfills, and if not properly disinfected, any pathogens present will be introduced to the crop being fertilized or ingested by wild animals at landfills (*Sano et al., 2003*). This study seeks to provide an extensive view of the circular ssDNA viruses present in a single WWTP to identify the viral families present in that environment along with the genetic diversity possible in those families.

## METHODS

### Viral isolation and DNA extraction

On February 5, 2012, 2.6 L of sewage were collected from the influent side of the aeration tank at Thomas P. Smith Wastewater Treatment Plant in Tallahassee, FL, USA. Immediately following collection, 60 mL of chloroform was added to destabilize any

plasma membranes present in the sample. The sample was then centrifuged for 15 min at $5000 \times g$. Supernatant was transferred to a clean flask, 72 g of NaCl was added, and the sample was incubated for 1 h at 4 °C. To further remove solids, the sample was centrifuged for 10 min at $10,000 \times g$. Supernatant was transferred to clean 500 mL bottles containing 39 g of PEG8000 and gently rocked to bring PEG into solution. Following an overnight incubation at 4 °C, the sample was centrifuged for 20 min at $10,000 \times g$ and the supernatant was discarded. The viral pellet was resuspended using 25 mL suspension media (0.58 g NaCl, 0.2 g $MgSO_4 \cdot 7H_2O$, 5 ml 1 M Tris pH 7.5, 0.1 g gelatin, 100 ml $H_2O$). The suspension was centrifuged for 5 min at $5,000 \times g$ to remove any remaining solids. The supernatant was concentrated using Amicon Ultra-15 centrifugal filter conicals (100,000 MWCO; EMD Millipore) until the total volume was between 500–600 μL. Sample was transferred to a clean tube and 100 μL of DNase and 100 μL DNase buffer (New England Biolabs) was added to remove any free-floating DNA. Sample was incubated for 30 min at 37 °C and then filter concentrated using Spin-X UF 500 filter concentrators (100,000 MWCO; Corning) until total volume was 500 μL. The entire sample was loaded onto a 5–30% sucrose gradient and centrifuged in an ultracentrifuge at 24,000 RPM for 110 min at 4 °C.

After centrifugation, 500 μL fractions were collected from the bottom of the tube, until the gradient was completely drained. An aliquot of 100 μL was reserved for plating and the remaining 400 μL had the protein degraded and the DNA extracted using a phenol chloroform DNA extraction method. In brief, the sample was heated to 95 °C for 15 min. After cooling to room temperature 10 μL of proteinase K (Invitrogen) was added to each fraction and incubated at 65 °C for one hour. Following protein degradation 1/10 volume 1:10 phenol:chloroform isoamyl alcohol (IAA) was added, each fraction was vortexed for 60–90 s and centrifuged in a bench top centrifuge for 5 min at $2000 \times g$. Supernatant was transferred to fresh tube, equal volume of 1:1 phenol:chloroform IAA was added and the previous vortex and centrifugation step was repeated. Supernatant was transferred to a clean tube and equal volume chloroform IAA was added; vortexing and centrifugation was repeated. A standard ethanol precipitation was performed by transferring the supernatant to a clean tube and adding 1/10 volume sodium acetate and $2\times$ volume 100% ethanol. Samples were inverted twice and stored overnight at −20 °C. DNA precipitation was completed by centrifuging samples for 20 min at $15,000 \times g$ at 4 °C. Supernatant was removed and 400 μL 70% ethanol was added to the precipitate and a final 5 min centrifugation at $15,000 \times g$ at 4 °C was conducted. Supernatants were removed and samples dried in a speedvac. The DNA was rehydrated in 50 μL $H_2O$.

## Determination of target fractions

WA13, a *Microviridae* that is easily cultured and maintained in a laboratory setting (*Rokyta et al., 2006*), was used in a control gradient to identify which fractions contained the target ssDNA genomes. The control gradient was performed using the protocol described above for the sample gradient. Each fraction of both gradients was spread on nutrient rich agar plates using *Escherichia coli* C as a host to determine which fractions

had the highest number of viable small bacteriophage. Every fraction from the sample gradient also had the DNA quantified on both a Nanodrop spectrophotometer and a QuBit (Life Technologies). The combination of these data was used to determine the six fractions to use for sequencing by correlating the WA13 containing fractions from the control gradient to the comparable DNA peak in the sample gradient.

## Amplification, library preparation, and sequencing

Rolling circle amplification of individual fractions was performed with a Genomiphi V2 kit (GE Healthcare) according to manufacturer's protocol. An ethanol precipitation was conducted on the amplified fractions as described above. Amplified fractions were quantified using a Qubit and aliquots of each amplified fraction were made containing at least 1 µg DNA; $H_2O$ was then added to a final volume of 200 µL. The fractions were sonicated in a Bioruptor (Diagenode) for 9 cycles of 30 s on high, 30 s off in order to fragment the DNA. Fractions were dehydrated and resuspended in 50 µL $H_2O$. Library preparation for a 300 cycle sequencing run was done using an Illumina TruSeq DNA Kit according to the manufacturers specifications. Individual fractions were multiplexed for identification after sequencing. Prepared samples were diluted to 10 nM and pooled together, final product was diluted to 2 nM. Sequencing was conducted on an Illumina MiSeq by loading 9 picomoles into the Illumina sequencing cartridge.

## Data analysis

Sequencing reads from the individual fractions were *de novo* assembled using SeqMan NGen (version 11) by DNAStar. All contigs greater than 2,000 nt long with at least 3X coverage were analyzed for circularity by looking for repeats at the beginning and end of the sequence. Circular contigs were considered full-genomes and were trimmed to remove duplicate ends. Full genome contigs were annotated using BLASTx for gene matches. Long open reading frames that did not get a hit on BLASTx were annotated as hypothetical genes. Pairwise comparisons were completed to remove duplicate genomes. For the determination of how many sequencing reads belonged to each viral group and to ensure accurate representation of single nucleotide polymorphisms (SNP) frequency, a second assembly was conducted using the annotated contigs as templates. The default parameters in SeqMan NGen for a metagenomic templated assembly, except no limitations were placed on deep regions was used. The SNPs were identified using SeqMan and considered valid when present in at least 20% of the mapped reads. The nonsynonymous (NS) SNPs were identified by pairwise gene comparisons between the consensus amino acid (AA) sequence and the resulting AA sequence with the SNP changes. Pairwise comparisons of the AA sequence of each gene to both their individual top hit from the BLASTx searches, and to all of the other members of their grouping in the community were performed to determine the percent identity in MegAlign by DNAStar. Cluster analysis was performed in R using the mclust package (*R Development Core Team, 2010*). Mclust was allowed to perform model selection to determine the best fit model and identify putative clusters (*Fraley & Raftery, 2002*). A genetic pairwise identity matrix was constructed for all individuals from each group for the capsid and

replication genes. Raw sequencing reads have been uploaded to the sequence read archive (SRR3580070). All of the genotypes have been uploaded to NCBI (*Circo/Geminiviridae*: KX259394–KX259454; *Microviridae*: KX259455–KX259476).

## RESULTS

### Community composition

All complete genomes with sequence similarities to the *Microviridae* family were most similar to the Gokushovirinae subfamily. The remaining whole genomes shared sequence similarities with either the *Circoviridae* or *Geminiviridae* families, or shared sequence similarities with both families, in the form of having one gene most closely related to the *Circoviridae* family and another gene most closely related to the *Geminiviridae* family. Due to the high level of recombination or intermediate forms in the *Circoviridae* and *Geminiviridae* families, all of the genomes that appeared similar to the *Circoviridae* or *Geminiviridae* families were combined together as one group (Fig. 1). Of the 7,723,150 total sequencing reads, 6,224,682 assembled into whole and partial genomes (whole: 5,118,276, partial: 1,106,406), leaving 1,498,468 that did not assemble, during a final templated assembly using the described genotypes as the references. The majority of whole genome (4,860,112 reads) and partial genome length contigs were members of the *Circo/Geminiviridae* group, accounting for 5,326,390 reads of sequencing assembled into 61 genotypes (Fig. 2). The remaining 22 whole genomes (258,164 reads) along with some of the partial genome length contigs were *Microviridae*, accounting for 319,493 of the sequences. Relatively few sequences were determined to be bacterial in origin (164,845 reads), and they were all short fragments of genomes. The remaining 413,954 sequences were either unknown, meaning that BLASTx searches did not provide any matches, or unclassified, indicating that the hits from BLASTx searches were to uploaded sequences of unknown origins and could be prokaryotic, eukaryotic, or viral. Of the 61 *Circo/Geminiviridae* genotypes, 32 contained two open reading frames (ORFs), 22 contained three ORFs, and seven had four ORFs. The capsid gene was annotated as hypothetical for 30 of the *Circo/Geminiviridae* genotypes as only the replication gene had a match on GenBank. Eighteen of the capsid genes were of *Circoviridae* ancestry, four were of *Geminiviridae* ancestry, and eight had top matches that were unclassified. The replication gene was annotated as hypothetical for one of the *Circo/Geminiviridae* genotypes as only the capsid gene had a match on GenBank. Thirty two of the replication genes were of *Circoviridae* ancestry, 12 were of *Geminiviridae* ancestry, and 16 had top matches that were unclassified. All of the *Microviridae* genotypes were most closely related to the Gokushovirinae subfamily for both the capsid and replication genes and did not contain any unexpected genes.

Pairwise comparisons were conducted for both the replication initiation (REP) and capsid gene AA sequences for each genotype against all of the other genotypes in their groupings to determine the percent identity between all of the genotypes in the sample. The majority of comparisons from the *Circo/Geminiviridae* group demonstrated that the pairs were less than 50% similar to each other, with very few isolated comparisons

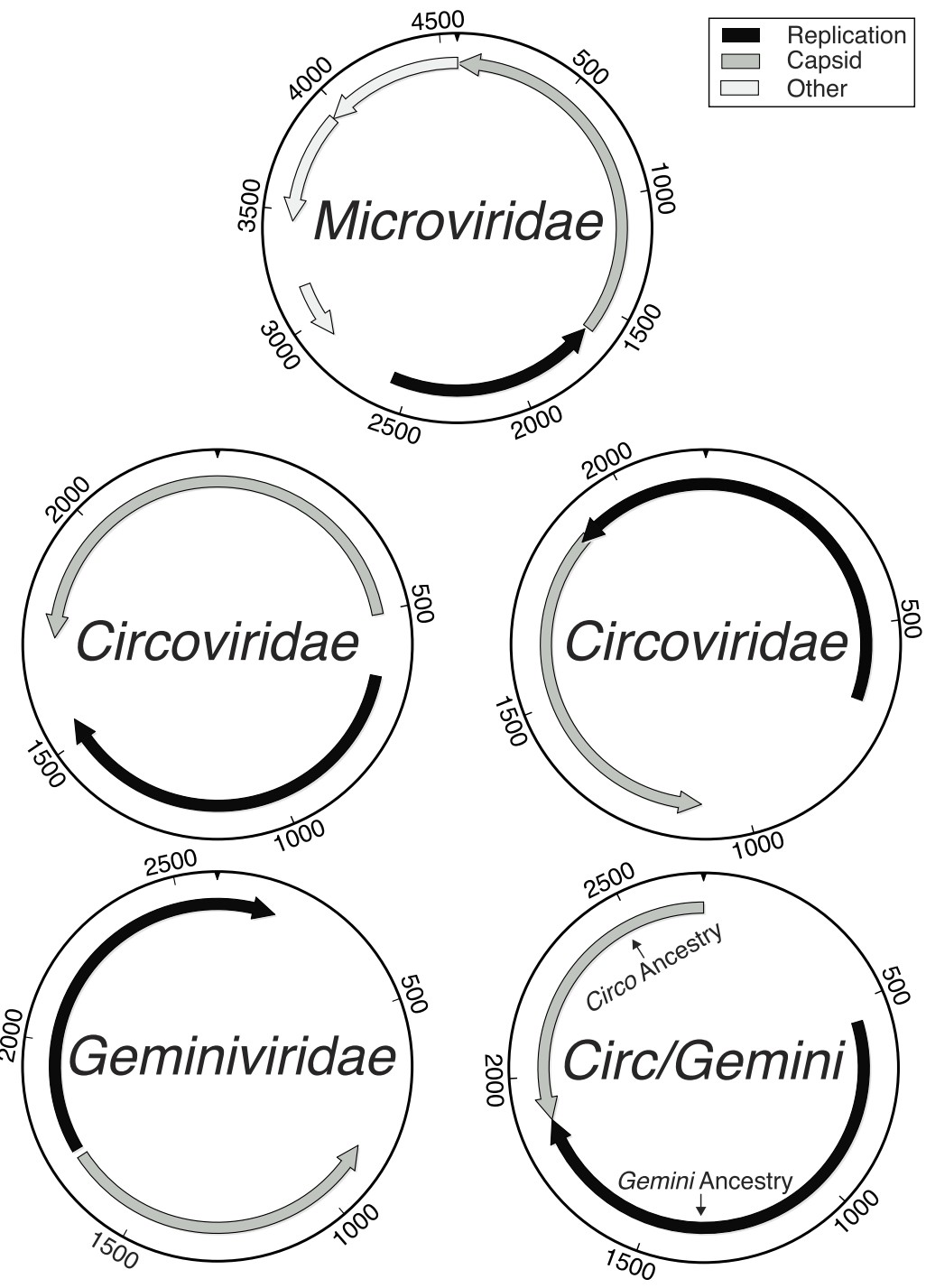

**Figure 1  Genome maps demonstrating the ambiguity of the *Circoviridae/Geminiviridae* genomes.**
Maps labeled as *Circoviridae* were examples of genotypes where both main genes had *Circoviridae* origins and were either uni- or bi-directional. All *Geminiviridae* genomes were bi-directional. The *Circ/Gemini* map displays an example of recombination, where each main gene has a different ancestral origin and the genomic organization is bi-directional. Although *Microviridae* have 9–11 genes, the novelty of the genotypes found limited the amount of genes that could be accurately annotated.

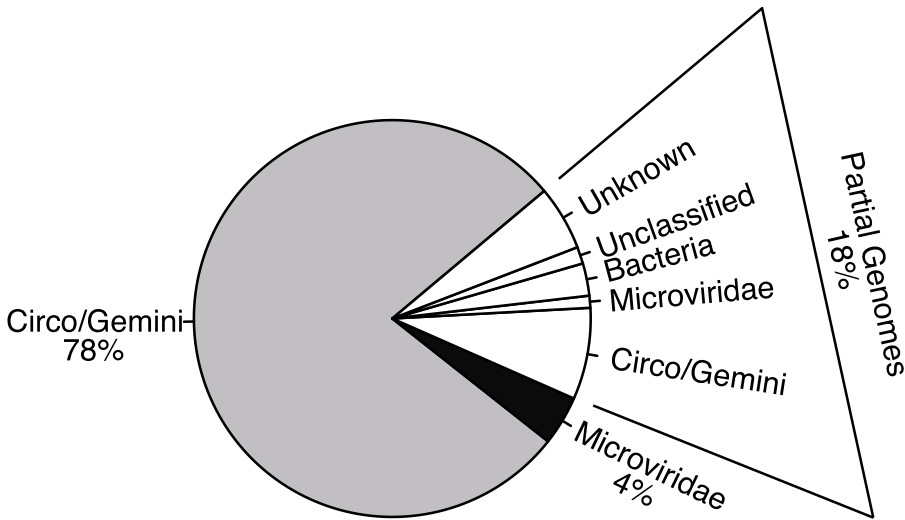

**Figure 2** **Distribution of sequencing effort demonstrates the efficacy of the protocol for isolating the ssDNA viruses.** Few sequences were recovered from bacterial contaminants and all were partial genomes (white). The majority of the sequences assembled into whole genomes from the combined *Circo/Geminiviridae* group (grey) and the *Microviridae* family (black) during a reference based assembly using the found genotypes as templates.

demonstrating a higher percent of identity (Fig. 3). The *Microviridae* group once again demonstrated a higher level of identity between members of their own community, however the majority of sequences were still below 60% identity. Cluster analysis identified nine clusters within the *Circo/Geminiviridae* (diagonal, equal shape model; BIC: −28847.64) group and seven clusters within the *Microviridae* (diagonal, varying volume and shape model; BIC: −3358.822) group (Fig. 3).

## Intercommunity diversity

Pairwise comparisons between the amino acid sequences for the REP and capsid genes from the recovered genomes and their top matches on BLASTx revealed that all of the recovered genes were <70% similar to their top hit (Fig. 4). Although all of the recovered genes only shared minimal AA identity with previously described viral genotypes, there were several that either had a common top match from BLASTx, or that shared top matches from the same study. Several members of the *Circo/Geminiviridae* group shared some AA sequence identity to genotypes discovered during other metagenomic studies in rodent stools (*Phan et al., 2011*), dragonflies (*Rosario et al., 2012*) and from seawater collected nearby in Tampa Bay, Florida (*Mcdaniel et al., 2013*). Interestingly, several members of the *Microviridae* group also shared top matches with genotypes discovered in dragonflies (*Rosario et al., 2012*). Multiple *Microviridae* group members also shared some AA sequence identity with genotypes discovered in ocean environments (*Labonté & Suttle, 2013a*). In the *Circo/Geminiviridae* group, it was common for one gene (in all but one case the REP gene) to have a match, but for the capsid gene to have no match on GenBank, resulting in many genomes having their capsid gene aligned at the 0% identity line. In the

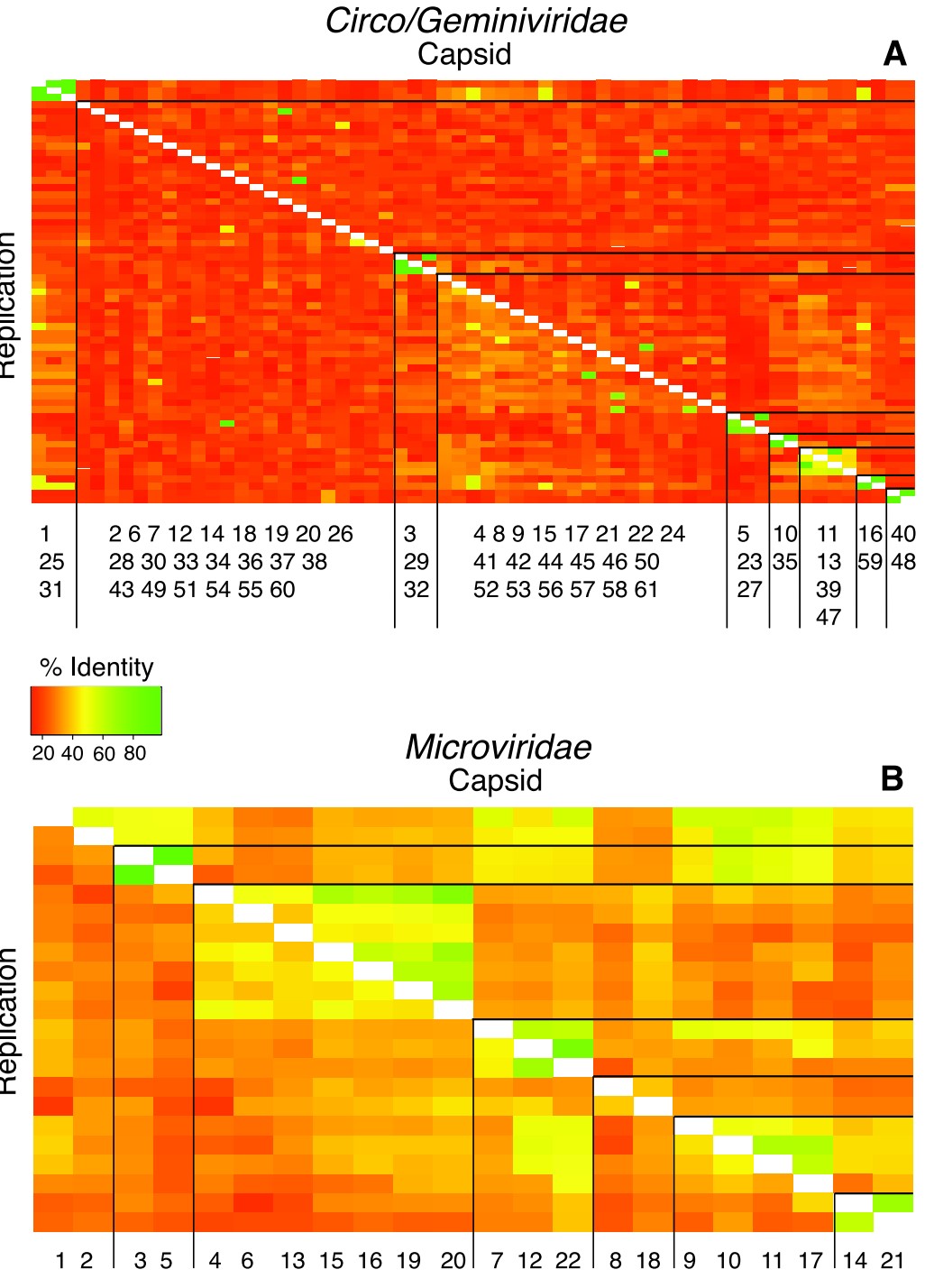

*Circo/Geminiviridae*
Capsid
**A**

% Identity

20 40 60 80

*Microviridae*
Capsid
**B**

**Figure 3** **Heat maps visualize pairwise comparisons for the capsid and replication genes AA sequences between each genotype in the population.** Very few genotypes within the WWTP are similar (more green), exhibiting a high level of dissimilarity (more orange). The genotypes are grouped based on cluster analysis into groupings of AA sequence identity using both capsid and REP genes. These groups may represent subfamilies and genera. The genotypes are listed along the *X*-axis within the cluster group in which they belong. The genotype numbers correspond to their official names on GenBank.

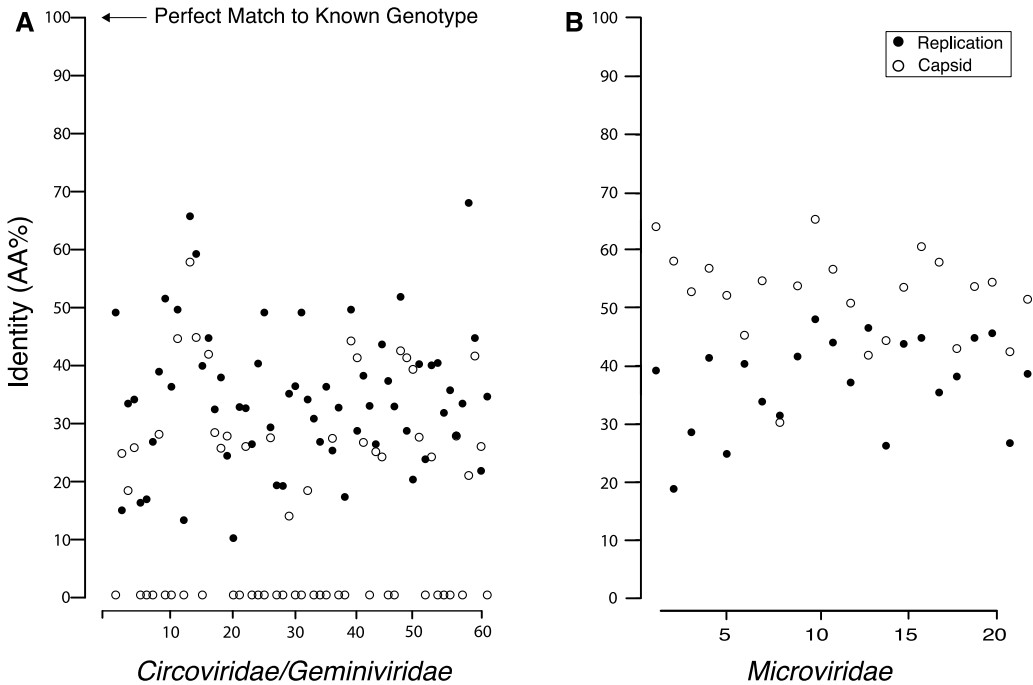

**Figure 4** **Percent identity between the capsid and replication genes for the discovered genotypes and their closest match from GenBank, which shows that all of the found genes are extremely dissimilar from their closest known relative.** Many of the *Circo/Geminiviridae* genotypes had one gene that did not have a match on GenBank and is therefore completely dissimilar to known viral genes.

*Microviridae* group both genes had a match on GenBank and all genes demonstrated at least 30% AA sequence identity.

## Genetic polymorphisms

The total number of SNPs within a genome were used to determine if the contig was an assembly of one or multiple potential genotypes. For both the *Circo/Geminiviridae* and *Microviridae* groups, there were a few contigs that did not have any SNPs above the reported cut-off and may be comprised of a single genotype. The remaining genomes contain multiple SNPs and can therefore be considered assemblages of closely related genotypes (Fig. 5). In the *Circo/Geminiviridae* group, half of the genotypes had between 0.0075–0.0113 SNP/nucleotide, whereas in the *Microviridae* group most of the genotypes had between 0–0.0038 SNP/nucleotide. For both the capsid and REP genes, the NS SNP frequency was determined (Fig. 6). Comparisons of the number of NS SNPs between the capsid and REP genes within both the *Circo/Geminiviridae* and *Microviridae* showed no significant differences (Welch's $t$-test, $p = 0.2$ and 0.5 respectively).

## DISCUSSION

We examined the diversity of ssDNA viruses in a wastewater treatment plant, by means of whole-genome sequencing. Previous studies have examined the phage diversity in WWTP using culture-based methods, which biased the results to only finding viruses that can be grown *in vitro* (*Rokyta et al., 2006*). Attempts to remove this bias have included shotgun

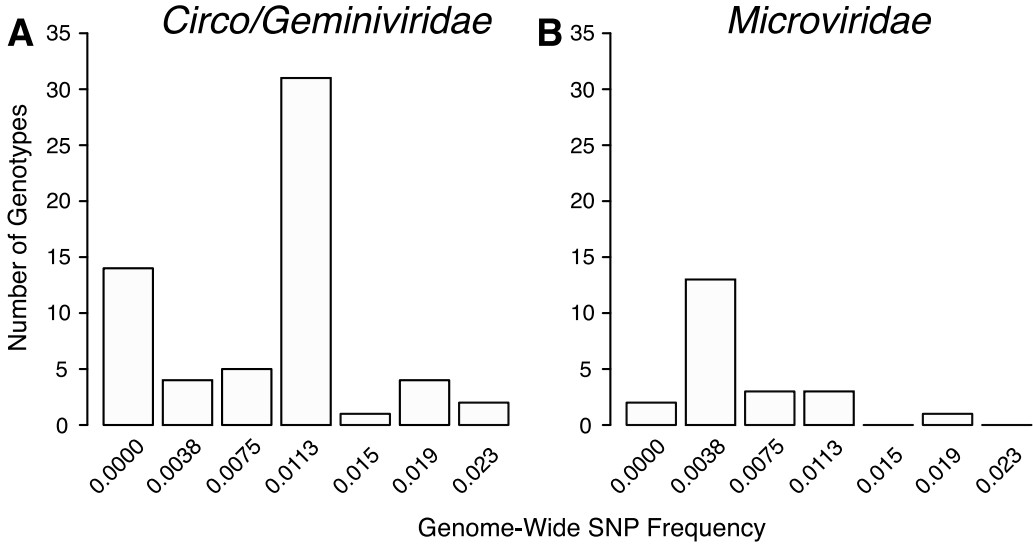

**Figure 5** **Distribution of SNP frequencies within identified genotypes.** Total number of genotypes containing SNPs at binned frequencies illustrates that many are actually genomic assemblages.

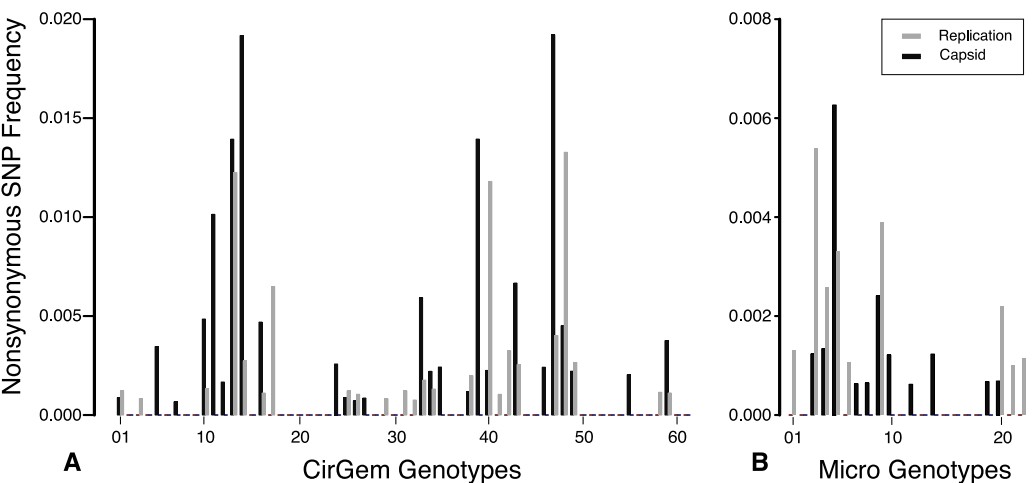

**Figure 6** **Nonsynonymous SNP frequency for the capsid and replication genes, demonstrating that the capsid and REP genes experience varying amounts of change in different genetic backgrounds.**

sequencing to get a rough description of the diversity, however, these studies mainly focused on the effluent leaving the treatment facilities or were biased towards dsDNA viruses (*Rosario et al., 2009*; *Parsley et al., 2010*; *Tamaki et al., 2012*). Other ssDNA virus diversity studies have focused on rice paddy soil and dragonflies using cloning techniques and Sanger sequencing (*Kim et al., 2008*; *Rosario et al., 2012*). Additional environmental studies reduced circular genome bias while investigating viromes in ocean water and human feces using 454 (*Kim et al., 2011*; *Labonté & Suttle, 2013b*). In this study, size-selection methods and amplification biases (*Kim & Bae, 2011*) minimized bacterial contamination and completely remove all large dsDNA and RNA viruses from
the sample, focusing the sequencing effort on ssDNA viruses. This method allowed for the ascertainment of information on all of the circular ssDNA viral families present in the system by removing the limitation of only recovering phage genomes.

Wastewater treatment plants rely on microorganisms to break down the organic matter present in the wastewater, and, as such, the viruses that infect these microbes are often abundant throughout the system (*Shapiro & Kushmaro, 2011*). Therefore, this study was initiated with the goal of investigating mostly *Microviridae* bacteriophage and potentially a few non-phage ssDNA viruses. By removing any culturing biases, we discovered a higher number of plant and animal virus genomes than bacteriophage, even though these are likely transient in the system. This conundrum of detecting more eukaryotic viruses than phage could represent bias of rolling circle amplification, because the *Circoviridae* and *Geminiviridae*-like viruses are much smaller than the *Microviridae* and should therefore amplify more quickly. Alternatively, it could be that the majority of ssDNA viruses are eukaryotic viruses, and this study uncovered an accurate portrayal of the percentage of ssDNA eukaryotic viruses versus prokaryotic viruses in the WWTP. Previous studies examining prokaryotic ssDNA viruses in WWTP have found that they are present in much lower numbers than the larger dsDNA viruses (*Rokyta et al., 2006*). Therefore, the genetic structure and limited gene composition of ssDNA viruses may make them better suited for infecting eukaryotic rather than prokaryotic hosts. The approach presented here has allowed a much broader taxonomic sampling than previously expected by facilitating characterization of the full genomic community of circular ssDNA viruses in the WWTP.

We found viral genomes with similarities to the known viral families *Circoviridae*, *Geminiviridae*, and *Microviridae*. However, for the *Circo/Geminiviridae* like genotypes, the AA sequence identity of both the capsid and REP gene, is <70% for all genotypes. Further, the AA sequence identity of all the capsid genes is <60%, and for many is <20%, indicating that they may all belong to new viral genera and subfamilies, demonstrated by the groupings assembled by cluster analysis (*Rosario, Duffy & Breitbart, 2012*). As the current known viral families on GenBank are primarily based upon viruses that have been discovered via standard culturing techniques, it is not surprising that multiple recent metagenomic studies have claimed the newly discovered genomes from their studies belong to new viral taxonomic groups of various levels (*Ng et al., 2009*; *Rosario & Breitbart, 2011*; *Phan et al., 2011*). More remarkable is that each metagenomic study has found new viral genera and families not present in the others, demonstrating the limited nature of the current understanding of viral taxonomic diversity (*Van den Brand et al., 2012*). Pairwise comparisons between all of the viruses identified determined that the majority are dissimilar from one another. Further, the SNP analysis showed that there are an unknown number of virus genomes assembled into the majority of genotypes in this study. Therefore, what this study refers to as genotypes may be groups of very similar viruses within these new taxonomic groups and more accurately be described as species rather than genotypes.

Several genotypes discovered were found to have one gene that was most closely related to one viral family (*Circoviridae*) while another gene was similar to a different viral family (*Geminiviridae*). We propose these to be recombination events, not relics of improper

assembly, as inspection of the assembly files showed level coverage across these genomes. Recombination events between both closely related viruses (*Rokyta et al., 2006*; *Lefeuvre et al., 2009*; *Roux et al., 2013*) and distantly related viral families has been reported with ssDNA viruses (*Diemer & Stedman, 2012*; *Krupovic et al., 2015*). As these genotypes are already dissimilar from the known viral genotypes within these families, it is unclear when these recombination events occurred. The recombined genotypes may represent continual recombination that is occurring between two closely related, but previously unknown viral groups. If this is the case, these new viruses potentially infect the same range of hosts, unlike the two viral families to which they are similar, which infect animals and plants, and may not have the opportunity for continual recombination. Conversely, this could be indicative of an ancient recombination event that occurred once or relatively few times, and diversification of the resulting recombinants led to the formation of these new viral groups that we are uncovering now. These recombinant genotypes may belong within either the *Circoviridae* or *Geminiviridae* families, or may represent a new viral family.

Wastewater treatment plants require microbes to properly digest and treat the influent, therefore viruses that infect these microbes are abundant in the system (*Sano et al., 2003*; *Parsley et al., 2010*; *Cantalupo et al., 2011*). The viral pathogens present do not aid in the digestion of the organic matter and may be hindering the digestion by interfering with the desired microbial populations (*Shapiro, Kushmaro & Brenner, 2009*). Only a small proportion of the recovered genotypes were closely related to known bacteriophage, however the majority of the recovered viral genotypes were too distantly related to any known genotypes as to be sure of their host range. The *Circoviridae* and *Geminiviridae* families are eukaryotic viruses that infect a broad spectrum of animals and plants respectively (*Bradeen, Timmermans & Messing, 1997*; *Cheung, 2012*; *King, Adams & Lefkowitz, 2012*). Therefore it is likely that our *Circo/Geminiviridae* like viruses are infecting eukaryotic hosts, but the host may not be the same as its closest match on GenBank. It is possible that the new viral groups are not merely washing into and out of the system, but rather are infecting unknown hosts that are permanently present within the system. Methods to elucidate the host of these viruses without culturing are needed to determine if these are transient viruses that infect plants and animals such as their distant relatives, or whether these new viruses are in fact residents of the WWTP. Alternatively, monitoring these viral populations could provide further support as to their stability in the system. Genotypes recovered across multiple time points could indicate permanent association with the WWTP; conversely limited sampling recovery provides support for transiency and minimal importance to the overall WWTP operation.

## ACKNOWLEDGEMENTS

We thank Margaret Seavy and Dr. Steven Miller for technical assistance throughout the processing of the sample, Dr. Lindsey McGee for providing guidance with SNP analysis, and Sarah Lueking and Kate Hill for providing writing guidance. We thank the employees of the Thomas P. Smith Water Reclamation Facility for assisting with sample collection.

### Funding

The authors received no funding for this work.

### Competing Interests

The authors declare there are no competing interests.

### Author Contributions

- Victoria M. Pearson conceived and designed the experiments, performed the experiments, analyzed the data, wrote the paper, prepared figures and/or tables, reviewed drafts of the paper.
- S. Brian Caudle analyzed the data, prepared figures and/or tables, reviewed drafts of the paper.
- Darin R. Rokyta conceived and designed the experiments, reviewed drafts of the paper.

### Data Availability

Raw sequencing reads were uploaded to the NCBI Sequence Read Archive under accession SRR3580070. All genotypes were uploaded to GenBank (*Circo/Geminiviridae*: KX259394–KX259454; *Microviridae*: KX259455–KX259476).

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
