# Peer review of "Viral recombination blurs taxonomic lines: examination of single-stranded DNA viruses in a wastewater treatment plant"

_PeerJ, doi:10.7717/peerj.2585_

## Round 0.1 · original submission · Minor Revisions

Please address the comments of both reviewers. You definitely need to use "recombinant" virus rather than hybrid virus and to cite and discuss the relevant work of others (reviewer 1). It would also be nice to have a feel for what percentage of genes in the assembled genomes have unknown functions and how many are appearing for the first time (e.g. reviewer 2). Several of your figures are clearly opaque to some readers (including me) and need much better explanation and labeling.

·

Basic reporting

The manuscript is well and clearly written.

The authors are missing some key information and references in the field to put their research into context. While the authors focused mainly on waste water treatment plants, I think it is important to add more references regarding what is known and has been done to know more about environmental ssDNA viruses. (See details below)

Figure 2 needs to be better describes and could be modified with the information from Table 1 for more clarity. (See details below)

Experimental design

There should be more connections done on what has been done in the past regarding environmental ssDNA viruses to better put the presented results in a global context. (See details below)

Validity of the findings

No comments.

Additional comments

In this manuscript, Pearson et al. present the results of a metagenomic analysis of ssDNA viruses in a waste water treatment plant. They enriched their sample in small circular ssDNA viruses by using a combination of size fractionation (with CsCl gradient) and whole genome amplification (which is biased towards small circular ssDNA viruses). They identified multiple new ssDNA viruses, including Gemini-Circoviridae hybrids.

Major comments:

Somehow, the term “hybridization” in the title is slightly misleading and they mean the creation of hybrids. My first thought was that they performed DNA-DNA hybridizations as part of the experiment. Better terms could be “recombination”, “hybrid formation”, or even “chimera formation”.

The authors fail to report major advancements in the study of environmental ssDNA viruses over the years and how their method compares to the ones previously used:
Kim et al., A&M, 2008
Kim et al., A&M, 2011
Labonté and Suttle, ISME J., 2013

In the light of other studies of ssDNA viruses, the authors should discuss if the ssDNA viruses they sequenced are similar to any of the environmental groups identified in previous studies (such as Labonté and Suttle, ISME J., 2013), even though waste water treatment plants are different than marine environments. Also, any similarity to cycloviruses, which were observed in chimpanzee stools (Li et al., 2010, Journal of virology).

Recombination in ssDNA viruses is not uncommon and the authors only mention one study where Gemini-Circoviruses were observed. There are, however, previous studies indicating a high incidence of recombination in ssDNA viruses (Lefeuvre et al., 2009, Journal of Virology). Moreover, recombination between ssDNA viruses and distantly related viruses has also been observed in:
Diemer and Stedman, 2012, Biology Direct
Krupovic et al., 2015, Genome Biol. Evol.

One discussion point that is missing is the about the putative origin of these viruses. They mention in the introduction that viruses in waste water treatment plants can come from various origins. Are the viruses that they observed more similar to viruses of human food (indirect input from human digestion), aquatic viruses (freshwater from street runoff), or other environments? The origin of these viruses is important in the aftermath of the waste water treatment and the importance of their possible further dissemination in the environment.

Specific comments:

Introduction:

Lines 74-80: When the authors discuss about the viral families, they did not mention all the Microviridae sub-families. Pichovirinae (Roux et al., 2012, PLoS One) and Alpavirinae (Krupovic and Forterre, 2011, PLoS One) are not discussed. Did the Microviridae identified in this study were similar to Pichovirinae or Alpavirinae?

Results:

Figure 1: Define the grey scale color scheme in the labeling of the arrows. For example, are the black arrows Rep?

Lines 181-187: In the Gemini-Circo hybrids, is there a trend showing that the replication proteins were more similar to the Circos and the capsid protein more similar to the Geminis, or vice versa?

Lines 183-192: State how many complete genomes were assembled for each family.

Figure 3: This figure is very confusing and the legend needs better explanations. Was any similarity clustering possible? Ordering the genomes by similarity clusters could make the figure easier to interpret, as well as demonstrating if the same groups are observed in Rep and Cap.

Table 1: Not sure what was the threshold and method used to determine the genotypes. Maybe this table could be combined with Figure 3 to order the genomes.

Figure 6: Is it possible to compute a statistics on the statement that there are more SNPs in the capsid in Circos and in the Rep for Micros.

Discussion:

Lines 237-242: There could be a bias due to the methodology in overamplifying eukaryotic viruses, but it would be interesting to discuss that maybe ssDNA viruses are just better adapted to infect eukaryotes.

Line 259: “Hybridization”, or more precisely “recombination”. Again, it is important to distinguish if the recombination occurred between a Rep for a Circo and a capsid from a Gemini, or between both. The Rep from both Gemini and Circo is different enough to be distinguished.

Reviewer 2 ·

Basic reporting

Overall, fine

Some issues to fix:
In Figure 1, I found the labels “circo origin” and “Gemini origin” a bit confusing. Looking at a circular DNA map, my first thought was that "origin" referred to the origin of replication.

I’m confused about figure 5. It’s not clear if the y-axis represents the number of closely-related genotypes for a given contig or whether it represents the number of different contigs that have a given percentage of SNPs.

To better contextualize the study, bring out the background on the ecology of these circular ssDNA viruses in waste water. The host ranges of the three families could be made more prominent, as could the expectation that the bacteriophage family would most prevalent in waste water.

The authors suggest that studies such as this one will make us "better equipped to handle future outbreaks." Please elaborate. I don't follow.

The writing is verbose and the sentence structure is sometimes convoluted. Please edit.

Experimental design

Seems fine, although it's not really my expertise.

Some issues to fix:

After sedimentation in a density gradient, various fractions were examined for viruses and DNA. Was anything learned about the sizes of viral particles in waste water?

Validity of the findings

It would be nice if the authors would comment on the gene content of the fully assembled genomes. How many genes were encoded in each assembled genome? Are there unexpected genes?

Additional comments

This study was interesting to read, and the data will likely be useful to other researchers.

---

## Round 0.2 · Minor Revisions

You frequently refer to "sequence similarity" without defining what you mean. From the text I deduce that you mean something about relationships at the amino acid level. This should be explicit everywhere.

It is also unclear whether you mean amino acid similarity or amino acid identity. I presume that you mean identity. If so, please state.

I am also unsure of why you think that 70% identity(?) is required for assignment to families, since at least some viral families may show as little at 22% amino acid sequence identity in some proteins.

---

## Round 0.3 · accepted · Accept

Thanks for your revision. We are now accepting your manuscript.